# SARS-CoV-2 and Autoimmune Cytopenia

**Ryann Quinn and Irina Murakhovskaya ***

Division of Hematology, Department of Hematology-Oncology, Montefiore Medical Center/Albert Einstein College of Medicine, Bronx, NY 10461, USA; ryquinn@montefiore.org
* Correspondence: imurakho@montefiore.org

**Abstract:** Severe acute respiratory syndrome coronavirus 2 (SARS-CoV-2) infection is associated with a variety of clinical manifestations related to viral tissue damage, as well as a virally induced immune response. Hyperstimulation of the immune system can serve as a trigger for autoimmunity. Several immune-mediated manifestations have been described in the course of SARS-CoV-2 infection. Immune thrombocytopenic purpura (ITP) and autoimmune hemolytic anemia (AIHA) are the most common hematologic autoimmune disorders seen in the course of SARS-CoV-2 infection. Vaccine-induced thrombocytopenia is a unique autoimmune hematologic cytopenia associated with SARS-CoV-2 vaccination. This paper will review the current literature on the association of SARS-CoV-2 infection and vaccination with autoimmune cytopenias and the clinical course of autoimmune cytopenias in patients with COVID-19.

**Keywords:** autoimmune; cytopenia; autoimmune hemolytic anemia; cold agglutinin syndrome; COVID-19; SARS-CoV-2; ITP; immune thrombocytopenia





## 1. Introduction

Since the first case of infection with novel SARS-CoV-2 virus was reported in Wuhan China in December of 2019, over 166 million people have been diagnosed and 3.4 million have died worldwide, as of 22 May 2021, and 1.6 billion doses of SARS-CoV-2 vaccines have been distributed [1]. This pandemic has had a devastating impact on human lives and has brought immense burden to the healthcare system. Symptoms of the infection range from asymptomatic infection, in at least 30% of cases [2], and mild respiratory illness, to severe complications, such as acute respiratory distress syndrome (ARDS) and multi-organ failure [3].

Activation of the immune system induced by SARS-CoV-2 can also contribute to the disease pathogenesis and organ damage. Immune hyperstimulation and dysregulation can lead to marked cytokine release syndrome, macrophage activation, and systemic hyperinflammation. Supporting the role of the inflammation-induced tissue damage, immunomodulatory and immunosuppressive agents have shown to have efficacy in the treatment of coronavirus disease 2019 (COVID-19), particularly in severe cases [4].

Hyperactivation of the immune system can trigger autoimmune manifestations in patients with prior history of autoimmune disease, as well as prompt the development of de novo autoimmune manifestations. Infections are an established risk factor for the development of autoimmune cytopenias. Viral infections linked to autoimmune hematologic disorders include the human immunodeficiency virus, hepatitis C virus and cytomegalovirus, parvovirus B19, Epstein–Barr virus, and Zika virus [5–7]. SARS-CoV-2 infection has been associated with several autoimmune complications, including cutaneous rashes and vasculitis, autoimmune cytopenia, anti-phospholipid syndrome, central and peripheral neuropathy, myositis, and myocarditis [8,9]. In a systematic review of 94 patients with COVID-19 who developed hematologic autoimmune disorders in their course of infection, the most common hematologic autoimmune disorder was immune thrombocytopenic purpura (ITP), seen in 58%, followed by autoimmune hemolytic anemia (AIHA) in 23% [10].

This paper will review the contemporary literature on the role of SARS-CoV-2 in the activation of the immune system and the association with autoimmune cytopenias, with a focus on autoimmune hemolytic anemia and ITP. It will also summarize available information on the association of SARS-CoV-2 vaccination and autoimmune cytopenias.

## 2. COVID-19 and Activation of the Immune System

Cytokine storm syndromes, such as ARDS and hemophagocytic lymphohistiocytosis (HLH), are seen in a subgroup of severely ill patients with COVID-19, supporting the role of the immune system dysregulation in the SARS-CoV-2 infection [11]. Both mild and severe forms of COVID-19 disease have been associated with changes in circulating leukocyte subsets and the upregulation of cytokine secretion, where marked increases in the secretion of cytokines of IL-6, IL-1β, IL-10, IL-17, TNF, and GM-CSF, referred to as 'cytokine storm', are seen in severe cases [12]. In addition, an imbalance in the Th17/Treg ratio and lower levels of regulatory T cells, which are involved in the downregulation of the immune response, have also been described in severe cases [13,14]. Mobilization of Th17 responses has been implicated in the pathogenesis of autoimmune diseases, including autoimmune hemolytic anemia [15,16], and low levels of regulatory CD4 T cells have been seen in patients with warm AIHA and ITP [17,18]. Lower levels of regulatory T cells in COVID-19 patients with severe infections can explain the higher rate of ITP in severe cases [19].

Multiple autoantibodies have been reported in association with SARS-CoV-2 infection, including antinuclear antibodies, cytoplasmic anti-neutrophil cytoplasmic antibodies, perinuclear anti-neutrophil cytoplasmic antibodies, anti-actin antibodies, and antimitochondrial antibodies [9]. High rates of lupus anticoagulant positivity rates associated with thrombosis, as well as anticardiolipin (aCL) and beta2 glycoprotein I (β2GPI), were reported in patients with COVID-19 infection [20,21]. Anti-heparin/platelet factor 4 (PF4) antibodies, which are associated with platelet activation in heparin-induced thrombocytopenia (HIT), have been recognized in severely ill COVID-19 patients with a HIT syndrome [22]. Some cases of HIT and positive PF4 antibodies have been reported without prior heparin exposure [23]. Antibodies to ADAMTS-13, a disintegrin and metalloproteinase with a thrombospondin type 1 motif, member 13, with a clinical syndrome of thrombotic thrombocytopenic purpura (TTP), were described as well [24,25].

Cross-reactivity between SARS-CoV-2 proteins and a variety of tissue antigens has been reported [26,27], supporting the role of molecular mimicry in the development of autoantibodies [28]. A structural similarity between ankyrin 1, a red blood cell membrane adaptor protein defective in patients with hereditary spherocytosis, and the SARS-CoV-2 surface glycoprotein, named the "spike protein", has been demonstrated [29]. A positive direct antiglobulin (DAT) test, which detects the presence of antibodies and or complement on the surface of red blood cells, has been reported in 13% of 267 anemic COVID-19 patients in one study, with higher rates of positivity in the ICU patients [30]. Another study reported a positive DAT in 46% of the 113 patients with COVID-19 [31].

In addition, greater than one-third of the immunogenic proteins in SARS-CoV-2 have been shown to have homology to proteins essential in the human adaptive immune system, providing support to the role of pathogenic priming in disease severity by induction of autoimmunity [32]. Induction of the autoimmune response against proteins in the adaptive immune system can impair major histocompatibility complex (MHC) class I and class II antigen presentation, cross-presentation of exogenous antigens, and programmed cell death protein 1 (PD-1) signaling [32], further contributing to immune dysregulation.

## 3. SARS-CoV-2 Infection in Patients with History of Autoimmune Disease

Patients with a history of autoimmune disease are potentially at risk for SARS-CoV-2 infection due to underlying immune dysregulation and use of immunosuppressive therapies. Alternatively, many immunomodulatory agents used in autoimmune diseases are utilized in the therapy of severe SARS-CoV-2 infection and can potentially curb the immune

response and associated organ damage. It appears that SARS-CoV-2 infection in patients with autoimmune diseases runs a generally mild course, comparable to that of the general population. A cross-sectional study of 916 patients with autoimmune rheumatologic disorders in Italy identified 148 symptomatic patients. Symptoms were typically mild and similar to the general population and no deaths were seen [33]. A case registry of 600 patients from 40 countries from the Global Rheumatology Alliance reported a more severe course, with 46% patients requiring hospitalization and 9% resulting in death [34]. In a prospective case series of 86 patients with a history of immune-mediated inflammatory diseases treated with anticytokine biologics and/or other immunomodulatory therapies who developed confirmed or highly suspected symptomatic SARS-CoV-2 infection, the incidence of hospitalization was 16%, with one case of mortality among 14 admitted patients. This was comparable to the incidence of hospitalization in the general population [35].

With regards to SARS-CoV-2 infection in autoimmune cytopenias, in a cohort of 501 patients in northern Italy with autoimmune cytopenias, which included 139 patients with warm AIHA (wAIHA), 108 patients with cold agglutinin disease (CAD), and 103 patients with ITP, four patients developed SARS-CoV-2 pneumonia. Among those, three patients had autoimmune hemolytic anemia (one CAD, one wAIHA, and one Evan's syndrome) and one had ITP. Relapse of hemolysis was seen in one patient with CAD and one with Evan's syndrome, and was successfully treated with transfusion and steroids [36]. An unfortunate case of recurrent warm autoimmune hemolytic anemia in the setting of COVID-19 infection complicated by lethal cryptococcal sepsis and encephalitis in the setting of immunosuppression with cyclophosphamide and steroids has been reported as well [37].

In the largest series of 32 chronic ITP patients who tested positive for SARS-CoV-2 a median of 39 months after ITP diagnosis, 56% of patients had moderate to severe COVID-19 disease requiring hospitalization and 38% of patients required non-invasive or invasive ventilation [38]. However, the authors note that cases of mild or asymptomatic SARS-CoV-2 infection may have been missed given that many patients during this time were followed via telemedicine and patients were only tested if they sought medical care for COVID-19 symptoms. A total of 47% of patients had a relapse of ITP requiring treatment an average of 9 days after SARS-CoV-2 diagnosis. The patients who required hospitalization had higher rates of ITP relapse than the patients who did not require hospitalization (74% vs. 15%, $p = 0.006$), suggesting that relapse is more common in patients with more severe COVID-19 symptoms. Most patients who required treatment received corticosteroids, and the two patients who did not respond to corticosteroids and intravenous immunoglobulin (IVIG) expired from respiratory failure from COVID-19 disease. Given that almost half of the patients who had SARS-CoV-2 experienced a relapse of ITP, clinicians should carefully monitor patients with a history of ITP who test positive for SARS-CoV-2 for bleeding symptoms. The mortality rate in this study was 9%, and death was related to SARS-CoV-2 severity rather than bleeding associated with ITP.

Interestingly, another smaller case series of eight patients with chronic ITP who had SARS-CoV-2 showed that four (50%) of the patients presented with early thrombocytosis and three patients required antiplatelet therapy, ITP treatment discontinuation, or ITP treatment reduction [39]. The authors speculate that lymphopenia could be responsible for the thrombocytosis, given that ITP is caused by auto-reactive B cells and altered regulatory T lymphocytes. Two patients (25%) had relapsed ITP and these patients showed a rapid response to treatment. Larger case series of patients with chronic ITP and subsequent SARS-CoV-2 infection are needed in order to clarify the association of thrombocytosis with SARS-CoV-2 infection.

## 4. SARS-CoV-2 as a Trigger for Development of Autoimmune Cytopenia

In the largest systematic review of 94 patients with SARS-CoV-2 who developed hematologic autoimmune disorder in the course of their infection [10], the most common hematologic autoimmune disorder was immune thrombocytopenic purpura, seen in 58%, followed by autoimmune hemolytic anemia in 23%. Other hematologic autoimmune

disorders included antiphospholipid syndrome (APLS) in 10 patients, TTP in 3 patients, Evans syndrome in 3 patients, and autoimmune neutropenia in 1 patient.

The time to onset of hematologic autoimmune disorders from the SARS-CoV-2 infection presentations was 11.8 ± 7.1 days. Among 22 patients with AIHA, cold agglutinin syndrome was seen in 8 patients. Three patients with AIHA complicating SARS-CoV-2 infection deceased.

### 4.1. SARS-CoV-2 and AIHA

To the best of our knowledge, 31 cases of AIHA have been reported presenting in the setting of SARS-CoV-2 infection, including 16 cases of wAIHA and 15 cases of cold agglutinin syndrome. Reports of wAIHA first presenting in the setting of theCOVID-19 infection are summarized in Table 1. Most cases of wAIHA were seen in adults, with three cases presenting in adolescents. The mean age in adults was 63.4 years old. Most cases presented within two weeks of the onset of the symptoms of SARS-CoV-2 infection. However, two patients had a delayed presentation 43 and 60 days into the course of the disease. One asymptomatic patient who tested positive for SARS-CoV-2 developed wAIHA 10 days after exposure. Three patients had a prior history of chronic lymphocytic leukemia (CLL) and one had monoclonal gammopathy of undetermined significance (MGUS) concurrently with wAIHA. One patient had a history of congenital thrombocytopenia and one had a history of ITP in association with immune dysregulation syndrome. Coombs was positive for IgG and complement in 8 out of 14 patients, IgA in 1 patient, and IgG only in 5 patients. Therapy included steroids and transfusion in most cases, and rituximab was administered in four patients, including two out of two patients with the presence of cold agglutinin. All patients recovered from SARS-CoV-2 and hemolysis was resolved in all cases with information available.

**Table 1.** WAIHA associated with SARS-CoV-2 infection.

| Age/ Gender | Comorbidity | DAT | Symptoms to AIHA Development, Days | AIHA Treatment | AIHA Outcome | COVID-19COVID-19 Outcome | Author |
|---|---|---|---|---|---|---|---|
| 14 Female | None | IgG, C3 CA | 0 | Steroids, Rituximab | Resolved | Recovered | Rosenzweig [40] |
| 13 Female | None | IgG | 7 | Steroids | Resolved | Recovered | Vega Hernádez [41] |
| 17 Male | ITP, ALPS | IgG, C3 | 0 | Steroids Transfusion | Resolved | Recovered | Wahlster [42] |
| 53 Female | Autoimmune thyroiditis | IgA | 60 | Steroids Transfusion Rituximab | Resolved | Recovered | Mausoleo [43] |
| 46 Female | Congenital thrombocytpenia | IgG, C3 | 3 | IVIG Transfusion Prednisone | Resolved | Recovered | Lopez [44] |
| 86 Female | HTN, anxiety, depression, MI | IgG | 43 | Steroids | Unknown | Recovered | Pelle [45] |
| 56 Male | HTN | IgG, C3 | 4 | IVIG Prednisone | Resolved | Recovered | Hindilerden [46] |
| 61 Male | CLL, CKD, HTN | IgG, C3 | 13 | Steroids | Unknown | Recovered | Lazarian [47] |
| 89 Female | CKD, HTN, MGUS | IgG, C3 | 7 | Steroids | Unknown | Recovered | Lazarian [47] |
| 75 Male | CLL | IgG | 6 | Transfusion | Unknown | Recovered | Lazarian [47] |
| 61 Male | DM | IgG | 9 | Steroids, Rituximab | Unknown | Recovered | Lazarian [47] |

**Table 1.** *Cont.*

| Age/ Gender | Comorbidity | DAT | Symptoms to AIHA Development, Days | AIHA Treatment | AIHA Outcome | COVID-19COVID-19 Outcome | Author |
|---|---|---|---|---|---|---|---|
| 33 Female | Iron deficiency anemia | IgG, C3 | 10 after exposure | Steroids Transfusion | Resolved | Asymptomatic | Liput [48] |
| 84 Male | Hyperlipidemia | IgG | 13 | Steroids Transfusion | Resolved | Recovered | Hsieh [49] |
| 33 Female | Hypothyroid | IgG, C3 CA | 2 | Steroids Transfusion tocilizumab Rituximab | Resolved | Recovered | Jacobs [50] |
| 72 Female | None | IgG | Unknown | Steroids Transfusion | Resolved | Recovered | Ramos -Ruperto [51] |
| 76 Female | HTN, CLL, hypothyroid | IgG | Unknown | Steroids Transfusion | Resolved | Recovered | Ramos -Ruperto [51] |

AIHA—autoimmune hemolytic anemia, ALPS—autoimmune lymphoproliferative syndrome, CA = cold agglutinin, C3—complement component 3, CKD—chronic kidney disease, CLL—chronic lymphocytic leukemia, DAT—direct antiglobulin test, DM—diabetes mellitis, ITP—immune thrombocytopenic purpura, HTN—hypertension, IgG—immunoglobulin G, MGUS—monoclonal gammopathy of undetermined significance, MI—myocardial infarction.

Table 2 summarizes reports of 15 patients who developed with cold agglutinin syndrome (CAS) in the setting of SARS-CoV-2 infection. The majority (nine) were males, and in five patients CAS was present at diagnosis. Hemolysis developed 5–20 days after the presentation in the rest of the patients. Four patients deceased as a consequence of SARS-CoV-2 infection and complications. In all other cases with available information, hemolysis resolved and patients recovered. Rituximab therapy [52] and plasma exchange were successfully used in one case each [51].

**Table 2.** CAS associated with SARS-CoV-2 infection.

| Age/ Gender | Comorbidity | Symptoms to AIHA Development, Days | AIHA Treatment | AIHA Outcome | COVID Outcome | Author |
|---|---|---|---|---|---|---|
| 62 Male | HTN, oropharyngeal squamous cell carcinoma on chemoradiation | 16 | Transfusion | Resolved | recovered | Capes [53] |
| 24 Female | None | 4 | None | Resolved | recovered | Moonla [54] |
| 51 Female | breast DCIS post-mastectomy on chemoradiation | 0 | Transfusion Steroids | Resolved | Recovered | Patil [55] |
| 48 Male | HTN, DM1, obesity ESRD on peritoneal dialysis | 7 | None | Unknown | Deceased concurrent DVT, stroke | Maslov [56] |
| 46 Female | ITP, asthma, splenectomy | 0 | Transfusion | Unknown | Deceased | Zagorski [57] |
| 80 Female | Stage A CLL | 12 | None | Resolved | recovered | Nesr [58] |
| 45 Male | Unknown | 0 | Transfusion | Unknown | Unknown | Raghuwanshi [59] |
| 77 Male | COPD, G6PD deficiency | 0 | Steroids | Unknown | Deceased | Gupta [60] |

**Table 2.** *Cont.*

| Age/ Gender | Comorbidity | Symptoms to AIHA Development, Days | AIHA Treatment | AIHA Outcome | COVID Outcome | Author |
|---|---|---|---|---|---|---|
| 61 Male | DM2, hypercholesterolemia, ESRD, CAD, atrial fibrillation | 5 | Steroids | Minimal hemolysis; Resolved | Recovered | Kaur [61] |
| 70 Male 67 Male | Unknown Unknown | 5 10 | None None | Minimal Hemolysis Minimal Hemolysis | Unknown Deceased | Jensen [62] |
| 43 Female 63 Male | untreated MS HTN | 16 20 | Transfusion Unknown | Recovered Recovered | Recovered Recovered | Huscenot [63] |
| 69 Female | Stage IV CLL on tirabrutinib, discontiniued | 18 | Steroids, Rituximab IVIG | Resolved | Recovered concurrent ITP, myositis | Aldaghlawi [52] |
| 54 Male | None | 0 | Steroids, Plasma exchange | Resolved | Recovered | Ramos-Ruperto [51] |

CAD—coronary artery disease, CLL—chronic lymphocytic leukemia, COPD—chronic obstructive pulmonary disease, DCIS—ductal carcinoma in situ, DM—diabetes mellitus, ESRD—end stage renal disease, G6PD—glucose-6-phosphate dehydrogenase, ITP—immune thrombocytopenic purpura, HTN—hypertension, IVIG—intravenous immunoglobulin, MS- multiple sclerosis, PE—pulmonary embolism, DVT— deep vein thrombosis.

### 4.2. SARS-CoV-2 and ITP

Immune thrombocytopenic purpura (ITP) has been described in numerous case reports and case series in association with SARS-CoV-2. The incidence of ITP in SARS-CoV-2 has been described in one study as 0.34% [64].

The majority of new cases of ITP in SARS-CoV-2 patients have been reported in patients 50–70 years old [10,19,64–68], although cases in children as young as 1 year and up to age 95 have been reported [64,69]. There have been two reported cases of pregnant women who developed de novo ITP associated with SARS-CoV-2 [64,70]. The increase in prevalence in older age is similar to ITP not associated with SARS-CoV-2 [71,72]. Unlike traditional ITP, SARS-CoV-2-associated ITP does not appear to be more common in younger females than in younger males, although the limited number of cases reported in patients younger than 50 might mask this association [10].

ITP associated with SARS-CoV-2 has been predominantly seen 7–14 days after the onset of symptoms, with many patients found to have thrombocytopenia on presentation to the hospital [10,19,64,68]. However, there have also been cases of delayed ITP occurring after one month of diagnosis [64], with 21% of cases in one series occurring after recovery from infection [19]. The variable time delay between COVID-19 symptoms and the occurrence of ITP in some cases may call into question the causal nature of SARS-CoV-2 with ITP. However, as some authors suggest, it could also reflect the diverse mechanisms of SARS-CoV-2 causing ITP [19,73].

The majority of patients in one series had moderate to severe SARS-CoV-2 [16], defined as the presence of dyspnea with evidence of pneumonia on radiology or lung auscultation [19]. A total of 36% of patients in another series had severe symptoms requiring intubation and all patients expired due to SARS-CoV-2 complications [64]. Seven patients (50%) had a World Health Organization (WHO) progression score of ≥5, which is consistent with SARS-CoV-2 requiring at least oxygen by nasal cannula [68,74].

However, ITP can occur in patients with mild symptoms, and in one study, 7% of ITP cases were in patients with asymptomatic COVID-19 [19]. This underscores the importance of ruling out SARS-CoV-2 in new ITP cases even if there are no COVID-19 symptoms. As the

pandemic continues, patients with newly diagnosed ITP should be tested for SARS-CoV-2 in addition to other classic viral triggers.

The average nadir platelet count in SARS-CoV-2-associated ITP is <10, similar to classical ITP [19,64,68]. Bleeding of any severity has been reported in 50–69% of patients; however, clinically significant bleeding is rare [19,64,68]. Most bleeding was cutaneous bleeding, minor gastrointestinal bleeding, hematuria, and epistaxis. There were few reports of ITP associated with intracranial hemorrhage. A case series of 45 patients found 4 patients to have intracranial hemorrhage (9%), with one mortality reported [19]. The bleeding risk in ITP associated with COVID-19 appears to be higher than previously reported in ITP not associated with COVID-19 [75]. The increase in bleeding could be due to the concomitant use of anticoagulation that many patients received due to the association of SARS-CoV-2 with coagulopathy.

Treatment of ITP associated with SARS-CoV-2 is similar to traditional ITP and most patients respond well to first line treatments (IVIG and corticosteroids). Many patients received initial treatment with IVIG alone or in combination with thrombopoietin receptor agonists (TPO-RA) due to concerns about corticosteroids worsening SARS-CoV-2 infection [19]. Additional patients did not receive corticosteroids due to concerns about diabetes and uncontrolled blood pressure. However, despite such concerns, the use of dexamethasone was not associated with worsened COVID-19 outcomes, and corticosteroids have shown to be associated with improved survival in COVID-19 patients [10,68,76]. In larger case series, response rates to ITP treatment have shown to be 70–90% [10,19,64,68]. This is consistent with outcomes seen in ITP not associated with SARS-CoV-2 [77]. In one study, among patients who received initial treatment with IVIG alone, 86% of patients responded [10]. A total of 84% of patients responded to single agent corticosteroids upfront. The median time to response was 4–10 days [64,68]. TPO-RAs have been used successfully and can decrease relapse rate [19]. Due to the potential thrombotic complications associated with TPO-RAs in patients that already have coagulopathy due to SARS-CoV-2, there have been recommendations to only use TPO-RAs as second line therapy for ITP in SARS-CoV-2 [78]. In one case series, TPO-RAs were used in nine patients for up to 3 weeks, and no thrombotic complications were reported [19]. Relapses have been reported in 9–21% of patients, although many patients in the reported cases did not have long follow up times [19,64,68].

In contrast to adults, there are very few case reports of children with ITP associated with SARS-CoV-2. This could be due to the fact that COVID-19 is less common in children than adults and that children generally have more mild cases with lower associated mortality [79]. In one series, only 7% of cases of SARS-CoV-2-associated ITP were in children <18 years old [19]. To our knowledge, there have been six reported cases of ITP in children with SARS-CoV-2 (Table 3). In the published cases in children, ITP has been associated with both mild and severe SARS-CoV-2 infection. ITP occurred in children from 17 months to 17 years of age. Two children were found to have ITP after recovering from multisystem inflammatory syndrome in children (MIS-C), which is a serious complication of SARS-CoV-2 presenting with fever, gastrointestinal symptoms, myocarditis, skin changes, and septic shock [69]. The treatment of MIS-C is similar to ITP and includes IVIG and high dose corticosteroids. In both patients, as corticosteroids for MIS-C were being tapered, they developed acute thrombocytopenia associated with minor mucosal bleeding and bruising. This suggests careful monitoring of platelets should be carried out as patients are being treated for SARS-CoV-2 and MIS-C. Similar to adults, there were two children who presented with ITP and were found to have asymptomatic SARS-CoV-2, which suggests that children with a new diagnosis of ITP for SARS-CoV-2, even in the absence of symptoms, should also be tested. Most children were treated with IVIG with or without steroids. All patients recovered from SARS-CoV-2 and there were no deaths. All patients responded to the treatment for ITP and no relapses were reported.

**Table 3.** Reported cases of SARS-CoV-2 -associated ITP in children.

| Author | Age (Years)/ Gender | Platelet Count Nadir | Bleeding | Severity of COVID Disease | Treatment | ITP Outcome | Days to Platelet Recovery |
|---|---|---|---|---|---|---|---|
| Patel [80] | 12/ female | <10 | Hematuria | Severe | IVIG Steroids | Recovered | 4 |
| Rosenzweig [40] | 16/ male | 4 | Petechiae/ purpura | Asymptomatic | Steroids | Recovered | 7 |
| Tsao [81] | 10/female | 5 | Petechiae/ purpura | Asymptomatic | IVIG | Recovered | 14 |
| Kok [69] | 17 months/male | 9 | Petechiae/ purpura | Severe | IVIG Steroids | Recovered | 4 |
| Kok [69] | 6/male | 45 | None | Severe | Steroids | Recovered | 9 |
| Soares [82] | 2/female | 16 | Petechiae/ ecchymosis | Asymptomatic | IVIG | Recovered | 5 |

IVIG—intravenous immunoglobulin.

### 4.3. COVID-19 Vaccination and Autoimmune Cytopenia

High immunogenicity of the spike protein led to it being a target in SARS-CoV-2 vaccine development. In total, 1.6 billion doses of SARS-CV-2 vaccines have been distributed as of 22 May 2022 [1]. A recent report of a healthy physician's death from the development of autoimmune thrombocytopenia and intracranial hemorrhage has raised the concerns of ITP as a complication of the SARS-CoV-2 vaccination [83].

A case series study of thrombocytopenia, including immune thrombocytopenia, after the receipt of mRNA COVID-19 vaccines reported to the Vaccine Adverse Event Reporting System (VAERS), reported a thrombocytopenia rate of 0.80 per million doses for Pfizer and Moderna vaccines. Based on an annual incidence rate of 3.3 ITP cases per 100,000 adults, the observed number of all thrombocytopenia cases, which included ITP, following administration of mRNA COVID-19 vaccines was not greater than the number of ITP cases expected [84]. A review of reports of 20 cases of thrombocytopenia following mRNA SARS-CoV-2 vaccination also suggests that the rate of newly diagnosed ITP is similar to the baseline incidence of ITP [85].

Vaccine-induced thrombocytopenia and thrombosis (VITT), in association with high titers of platelet-activating anti-platelet factor 4 antibodies, thrombocytopenia, cerebral venous sinus, and visceral thrombosis, have been described with vaccines containing replication-incompetent adenoviral vectors encoding the SARS-CoV-2 spike protein antigen (human [Ad26.COV2.S] for Janssen and chimpanzee [ChAdOx1] for AstraZeneca) [86–89]. Of the >200 million doses of mRNA-based vaccines administered in the US to date, there have been no documented reports of thrombosis complicated by thrombocytopenia [86].

Immunocompromised people, including those on immunosuppressive or immunomodulatory agents, immunoglobulin or blood products, asplenia, and autoimmune conditions such as immune thrombocytopenic purpura, were excluded from landmark phase 3 randomized controlled trials. The data on the effect of SARS-CoV-2 vaccination in this population are limited. The development of hemolytic crisis due to COVID-19 vaccination has been reported in a patient with cold agglutinin disease [90]. A prospective study of 52 chronic ITP patients reported a median platelet count drop of 96% in 12% of patients 2–5 days after vaccination, with new bleeding symptoms requiring therapy [91]. Current guidelines for the management of patients with ITP maintain that the risks associated with COVID-19 outweigh the risks associated with SARS-CoV-2 vaccination and suggest monitoring baseline and post-vaccination platelet counts in ITP patients, particularly if they are persistently thrombocytopenic or have a history of unstable platelet counts [92].

## 5. Discussion

Viral infections have been established as a risk factor for the development of autoimmune cytopenia. In SARS-CoV-2 infection, the inflammatory response leads to hyperstimulation of the immune system, which can promote the development of autoimmune complications in the susceptible individuals. Recognizing autoimmune cytopenia in COVID-19 is important in order to avoid a delay in treatment. Patients with a more severe clinical course are more likely to have lower hemoglobin and platelet levels, and there is a strong association between anemia and thrombocytopenia with poor outcome/mortality in hospitalized patients with SARS-CoV-2 [30,93].

Homology between the ankyrin-1 and the SARS-CoV-2 spike glycoprotein can lead to the development of red cell autoantibodies and potentially autoimmune hemolytic anemia in response to viral infection or vaccination. This can account for high rates of DAT positivity in patients with SARS-CoV-2 infection. It is important to note that a positive direct anti-globulin test can be seen in the absence of changes in the hemolytic parameters in healthy adults as well as hospitalized patients [15], and, in those cases, is not associated with clinical syndrome of autoimmune hemolytic anemia. A similar homology between SARS-CoV-2 and platelet antigens has not yet been reported.

Most cases of wAIHA that developed in the setting of SARS-CoV-2 infection were generally self-limited and responded to steroids and supportive measures. Rituximab administration was utilized in less than half of the patients. One mortality seen in a case of recurrent warm autoimmune hemolytic anemia triggered by SARS-CoV-2 infection was due to opportunistic infection in the setting of immunosuppression with cyclophosphamide and steroids. Generally favorable outcomes in patients with wAIHA suggest the importance of limiting the use of cytotoxic agents in treating autoimmune cytopenias in SARS-CoV-2 in favor of steroids and rituximab in refractory cases.

Cold agglutinin syndrome was seen in approximately 50% of the cases of AIHA in association with SARS-CoV-2 reported in the literature, compared to the reported overall 20–25% incidence in idiopathic AIHA [94]. In contrast, most cases of infection-related AIHAs are usually cold, IgM-mediated, and self-limited. Four out of fifteen patients with cold agglutinin syndrome in the setting of COVID 19 deceased as a result of SARS-CoV-2 infection, suggesting a higher mortality rate in this population. Complement activation plays a key role in the pathophysiology of cold autoimmune hemolytic anemia. The crucial and detrimental role of complement activation and preliminary efficacy of complement inhibition has been previously described in SARS-CoV-2 infection [95]. The presence of cold agglutinin syndrome can serve as a potential marker of adverse outcomes in SARS-CoV-2 infection and of consideration for immunomodulatory and/or complement directed therapy. Complement involvement was seen in almost two thirds of the reported cases of wAIHA in the setting of SARS-CoV-2 infection, which is much higher compared to primary wAIHA cases, with a large retrospective study reporting 30% of cases positive for IgG and complement [96]. However, clinical outcomes were much more favorable in wAIHA. Therapy was required more frequently in COVID-19-associated wAIHA compared to COVID-19-associated CAS, which is consistent with the need for therapy in idiopathic forms [17].

ITP associated with SARS-CoV-2 is more common in older adults and is infrequently seen in children. Thrombocytopenia typically occurs after 1 week of symptoms. Most patients with ITP are seen in patients with moderate to severe infection requiring oxygen. However, a significant portion can develop ITP with asymptomatic SARS-CoV-2 infection, suggesting that as the pandemic continues, the evaluation for SARS-CoV-2 should be included in the initial evaluation for viral etiologies for new onset immune thrombocytopenia. Bleeding might be more common than in ITP not associated with SARS-CoV-2; however, this could be due to anticoagulation used for coagulopathy, and larger studies need to be carried out. Similar to wAIHA, most cases of ITP associated with SARS-CoV-2 respond well to first line therapies such as steroids and IVIG.

Given that, by definition, ITP is a diagnosis of exclusion, other causes of viral-associated thrombocytopenia must be ruled out, such as disseminated intravascular coagulation (DIC), sepsis-induced bone marrow suppression, thrombotic microangiopathy, or drug-induced thrombocytopenia. Anti-platelet antibodies are not sensitive and cannot reliably be used to confirm the diagnosis of ITP. This becomes especially important in patients that are refractory to treatment; one must reconsider the diagnosis of ITP and look to other causes of thrombocytopenia.

In patients with a prior history of autoimmune cytopenias, the course of SARS-CoV-2 infection is generally favorable, and relapses of cytopenias are responsive to therapy. The baseline use of biologics and/or other immunomodulatory therapies was not associated with worse COVID-19 outcomes in patients with rheumatologic diseases [35]. On the contrary, it appears that patients who had immunosuppressive therapy discontinued in the setting of SARS-CoV-2 infection were more likely to develop autoimmune cytopenia relapse [36,52].

Vaccine administration has been associated with autoimmune manifestations in some genetically predisposed individuals; however, it has been demonstrated that vaccinations do not pose a more prominent danger than actual infections. Vaccine-induced immune thrombotic thrombocytopenia has been described in association with SARS-CoV-2 vaccines containing replication-incompetent adenoviral vectors that share similar clinical and laboratory features with autoimmune HIT [89,97]. Recommendations for clinical and laboratory diagnosis and management by the International Society of Thrombosis and Hemostasis (ISTH) have recently been published [98]. This rare complication of SARS-CoV-2 vaccination has an estimated incidence of ~7–10 cases per million individuals with the AstraZeneca vaccine and ~3.2 per million for the Johnson & Johnson vaccine as of 5/12/21, and has not been observed with mRNA vaccines [89].

The incidence of ITP does not appear to be increased following vaccination in large studies; however, relapses of chronic ITP are common. Complete blood count monitoring should be considered in patients with a history of autoimmune cytopenias at prior to and one week following SARS-CoV-2 vaccination.

## 6. Conclusions

SARS-CoV-2 infection is associated with the development of autoimmune cytopenias and should be included in the initial evaluation of patients with a new presentation or exacerbation of autoimmune hematologic disorders. The monitoring of post-vaccination blood counts is recommended in patients with pre-existing autoimmune cytopenias.

**Funding:** This research received no external funding.

**Institutional Review Board Statement:** Not applicable.

**Informed Consent Statement:** Not applicable.

**Data Availability Statement:** Not applicable.

**Conflicts of Interest:** RQ declares no conflict of interest. IM: research support from Alexion, Incyte Corporation, Kezar, Momenta/Janssen, Rigel, and Sanofi. Consultancy: Apellis, Momenta/Janssen, Novartis, Sanofi, Medscape.

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
