# Peer review of "SARS-CoV-2 and Autoimmune Cytopenia"

_hemato, doi:10.3390/hemato2030029_

Round 1
Reviewer 1 Report
The authors provide an updated review on the association between Covid-19 infection and autoimmune cytopenia, considering the occurrence of Covid-19 infection in patients with pre-existing autoimmune cytopenias as well as newly diagnosed autoimmune cytopenias complicating Covid-19 infection. The emphasis is on warm and cold AIHA and ITP, but vaccine induced thrombocytopenia and thrombosis (VITT) is also reviewed. The paper is comprehensive, systematic and well-written, and the literature references are adequate. This is an excellent article, and I have virtually no comments.
Minor comment: Please check that all abbreviations/acronyms are explained the first time they occur in the text (As far as I can see, HLH [line 57] has not been explained).
Author Response
We thank the referee 1 for the kind remarks. The paper has been perused and all abbreviations have been explained including line 57.
Reviewer 2 Report
This review is a comprehensive and well-written paper encompassing the available evidence on COVID-19 and autoimmune cytopenia (both in patients with pre-existing diagnosis and in new-onset cytopenic patients). Tables are well-organised and all topics are well discussed. I have just some minor comments which may add completeness in the discussion section:
1) it is worth noting that COVID19-associated warm AIHA required therapy more frequently than COVID19-associated CAS, which is in line with idiopathic forms.
2) differently from primary AIHA, the proportion of warm and cold AIHA cases n COVID19-associated forms were comparable. Besides COVID19, other infection-related AIHAs are usually cold IgM-mediated, self-limiting cases.
3) Thrombocytopenia in the context of a systemic infection is a multifactorial phenomenon, and the clinician has many confounders to take in consideration in the differential diagnosis: sepsis-induced thrombocytopenia, thrombotic microangiopathy, disseminated intravascular coagulopathy, thrombocytopenia due to splenic consumption, or to inflammatory cytokine-mediated bone marrow inhibition. Diagnosis of ITP in this context is difficult and, by definition, of exclusion. Additionally, the specificity of anti-platelet antibodies test is not optimal, especially in the context of acute inflammation/immune activation. Therefore, a comment about this point might be useful in the discussion.
4) For the same reason as above, it might be worth reminding that direct anti-globulin test (DAT) positivity does not necessarily mean AIHA diagnosis, since isolated DAT positivity in hospitalized patients can be present without alteration of hemolytic markers, making the diagnosis of AIHA unlikely. For the sake of completeness, this should be pointed out as a useful clinical hint in the discussion.
Author Response
We thank the Referee for the careful revision and for the important remarks. Discussion section has been expanded as suggested.
1) it is worth noting that COVID19-associated warm AIHA required therapy more frequently than COVID19-associated CAS, which is in line with idiopathic forms.
Discussion section has been addended to highlight more frequent need for therapy in COVID19-associated wAIHA compared to COVID-19-associated CAS which is consistent with idiopathic forms. We added a sentence lines 370-372.
2) differently from primary AIHA, the proportion of warm and cold AIHA cases n COVID19-associated forms were comparable. Besides COVID19, other infection-related AIHAs are usually cold IgM-mediated, self-limiting cases.
Proportion of warm and cold AIHA cases n COVID19-associated forms has been discussed in lines 352-354. We added a discussion point regarding most infection-related AIHAs being usually self-limited cold IgM-mediated cases lines 358-359.
3) Thrombocytopenia in the context of a systemic infection is a multifactorial phenomenon, and the clinician has many confounders to take in consideration in the differential diagnosis: sepsis-induced thrombocytopenia, thrombotic microangiopathy, disseminated intravascular coagulopathy, thrombocytopenia due to splenic consumption, or to inflammatory cytokine-mediated bone marrow inhibition. Diagnosis of ITP in this context is difficult and, by definition, of exclusion. Additionally, the specificity of anti-platelet antibodies test is not optimal, especially in the context of acute inflammation/immune activation. Therefore, a comment about this point might be useful in the discussion.
Thank you to referee 2 for the helpful comments. We agree that the diagnosis of ITP difficult especially in the context viral infection. We have added a comment about this in the discussion lines 378-384.
4) For the same reason as above, it might be worth reminding that direct anti-globulin test (DAT) positivity does not necessarily mean AIHA diagnosis, since isolated DAT positivity in hospitalized patients can be present without alteration of hemolytic markers, making the diagnosis of AIHA unlikely. For the sake of completeness, this should be pointed out as a useful clinical hint in the discussion.
We thank the referee for pointing this out. We made a distinction between DAT positivity in the absence of abnormalities in hemolytic markers and autoimmune hemolytic anemia in lines 344-347.